# Efficiently Identifying Watermarked Segments in Mixed-Source Texts

**Xuandong Zhao**[*]
UC Berkeley
xuandongzhao@berkeley.edu

**Chenwen Liao**[*]
Zhejiang University
liaochenwen@zju.edu.cn

**Yu-Xiang Wang**[†]
UC San Diego
yuxiangw@ucsd.edu

**Lei Li**[†]
Carnegie Mellon University
leili@cs.cmu.edu

## Abstract

Text watermarks in large language models (LLMs) are increasingly used to detect synthetic text, mitigating misuse cases like fake news and academic dishonesty. While existing watermarking detection techniques primarily focus on classifying entire documents as watermarked or not, they often neglect the common scenario of identifying individual watermark segments within longer, mixed-source documents. Drawing inspiration from plagiarism detection systems, we propose two novel methods for partial watermark detection. First, we develop a geometry cover detection framework aimed at determining whether there is a watermark segment in long text. Second, we introduce an adaptive online learning algorithm to pinpoint the precise location of watermark segments within the text. Evaluated on three popular watermarking techniques (KGW-Watermark, Unigram-Watermark, and Gumbel-Watermark), our approach achieves high accuracy, significantly outperforming baseline methods. Moreover, our framework is adaptable to other watermarking techniques, offering new insights for precise watermark detection.

## 1 Introduction

Large Language Models (LLMs) have revolutionized human activities, enabling applications ranging from chatbots (OpenAI, 2022) to medical diagnostics (Google, 2024) and robotics (Ahn et al., 2024). Their ease of use, however, presents serious societal challenges. In education (Intelligent, 2024), students can effortlessly generate essays and homework answers, undermining academic integrity. In journalism (Blum, 2024), distinguishing credible news from fabricated content erodes public trust. The potential for malicious uses, such as phishing (Violino, 2023), and the risk of model collapse due to synthetic data (Shumailov et al., 2024), further underscore the urgent need to detect LLM-generated text and promote the responsible use of this powerful technology.

However, identifying AI-generated text is becoming increasingly difficult as LLMs reach human-like proficiency in various tasks. One line of research (OpenAI, 2023; Tian, 2023; Mitchell et al., 2023) trains machine learning models as AI detectors by collecting datasets consisting of both human and LLM-generated texts. Unfortunately, these approaches are often fragile (Shi et al., 2024) and error-prone (Liang et al., 2023), ultimately leading OpenAI to terminate its deployed detector (Kelly, 2023). Watermarking has emerged as a promising solution to this challenge. By embedding identifiable patterns or markers within the generated text, watermarks can signal whether a piece of text originates from an LLM.

Existing watermark detection methods (Aaronson, 2023; Kirchenbauer et al., 2023; Zhao et al., 2023; Kuditipudi et al., 2023; Christ et al., 2023; Hu et al., 2024) are primarily designed for text-level classification, labeling a piece of text as either watermarked or not. However, these methods are insufficient for many real-world scenarios where documents contain mixed-source texts, and only specific sections are LLM-generated. For instance, malicious actors might use LLMs to manipulate

---

[*]Co-first authors.
[†]Co-supervised project.

certain sections of a news article to spread misinformation. Detecting watermarks within long, mixed-source texts presents a significant challenge, especially when aiming for subsequence-level detection with uncertainty quantification, similar to plagiarism detection systems like "Turnitin[1]". This is because the watermarked signal may be weakened throughout the increasing text length and may not be easily identifiable using conventional detection methods.

To bridge the gap, we propose partial watermark detection methods that offer a reliable solution for identifying watermark segments in long texts. A straightforward approach, which involves examining all possible segments of a text containing $n$ tokens, yields an inefficiently high time complexity of $\mathcal{O}(n^2)$. Instead, we employ the *geometric cover* trick (Daniely et al., 2015) to partition the long texts into subsequences of varying lengths and then perform watermark detection within each interval. This approach, termed the *Geometric Cover Detector* (GCD), enables efficient classification of whether a document contains any watermarked text in $\mathcal{O}(n \log n)$ time. However, GCD does not assign a score to every token, providing only a rough localization of watermark segments.

To refine this localization, we introduce the *Adaptive Online Locator* (AOL). AOL reformulate the problem as an online denoising task, where each token score from the watermark detector serves as a noisy observation for the mean value of scores within watermark segments. By applying an adaptive online learning method, specifically the *Alligator* algorithm (Baby et al., 2021), we retain the $\mathcal{O}(n \log n)$ time complexity while significantly improving the accuracy of detected segments.

We validate GCD and AOL using the C4 (Raffel et al., 2020) and Arxiv (Cohan et al., 2018) datasets, employing Llama (Touvron et al., 2023) and Mistral (Jiang et al., 2023) models for evaluation. Our empirical results demonstrate strong performance across both classification and localization tasks. In the classification task, our method consistently achieves a higher true positive rate compared to the baseline at the same false positive rate. For localization, we achieve an average intersection over union (IoU) score of over 0.55, far exceeding baseline methods.

In summary, our contributions are threefold:

1. We introduce novel approaches to watermark detection, moving beyond simple text-level classification to identification of watermark segments within long, mixed-source texts.
2. We employ the *geometric cover* trick and the *Alligator* algorithm from online learning to reliably detect and localize watermark segments efficiently and accurately.
3. We conduct extensive experiments on state-of-the-art public LLMs and diverse datasets. Our empirical results show that our approach significantly outperforms baseline methods.

## 2 BACKGROUND AND RELATED WORK

**Language Models and Watermarking.** A language model $\mathcal{M}$ is a statistical model that generates natural language text based on a preceding context. Given an input sequence $x$ (prompt) and previous output $y_{<t} = (y_1, \ldots, y_{t-1})$, an autoregressive language model computes the probability distribution $P_{\mathcal{M}}(\cdot|x, y_{<t})$ of the next token $y_t$ in the vocabulary $\mathcal{V}$. The full response is generated by iteratively sampling $y_t$ from this distribution until a maximum length is reached or an end-token is generated. *Decoding-based watermarking* (Aaronson, 2023; Kirchenbauer et al., 2023; Zhao et al., 2023; Kuditipudi et al., 2023; Christ et al., 2023; Hu et al., 2024) modifies this text generation process by using a secret key k to transform the original next-token distribution $P_{\mathcal{M}}(\cdot|x, y_{<t})$ into a new distribution. This new distribution is used to generate watermarked text containing an embedded watermark signal. The watermark detection algorithm then identifies this signal within a suspect text using the same watermark key k.

**Red-Green Watermark.** Red-Green (statistical) watermarking methods partition the vocabulary into two sets, "green" and "red", using a pseudorandom function $R(h, k, \gamma)$. This function takes as input the length of the preceding token sequence ($h$), a secret watermark key (k), and the target proportion of green tokens ($\gamma$). During text generation, the logits of green tokens are subtly increased by a small value $\delta$, resulting in a higher proportion of green tokens in the watermarked text compared to non-watermarked text. Two prominent Red-Green watermarking methods are KGW-Watermark (Kirchenbauer et al., 2023; 2024) and Unigram-Watermark (Zhao et al., 2023). KGW-Watermark

---

[1]https://www.turnitin.com

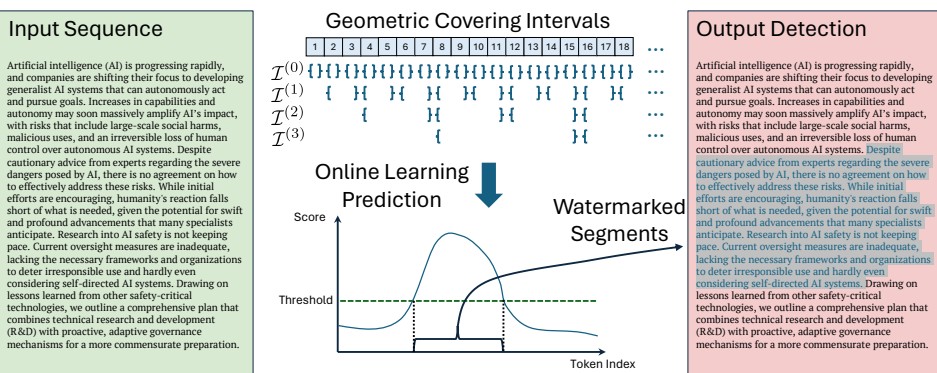

Figure 1: Illustration of the watermark segment detection process. The input sequence could be mixed-source of watermark text and unwatermark text. The input sequence could be a mixed-source of watermarked text and unwatermarked text. We use geometric covers to partition the text and detect watermarks in intervals. We also formulate localization as an online denoising problem to reduce computational complexity. The example shown is drawn from the abstract of Bengio et al. (2024), with the watermarked part generated by a watermarked Mistral-7B model.

utilizes $h \geq 1$, considering the prefix for hashing. Unigram-Watermark employs fixed green and red lists, disregarding previous tokens by effectively setting $h = 0$ to enhance robustness. Watermark detection in both methods involves identifying each token's membership in the green or red list

$$\text{Score}(y) = \sum_{t=1}^{n} \mathbf{1}(y_t \in \text{Green Tokens}) \tag{1}$$

and calculating the $z$-score of the entire sequence:$z_y = \frac{\text{Score}(y) - \gamma n}{\sqrt{n\gamma(1-\gamma)}}$. This $z$-score reflects the deviation of the observed proportion of green tokens from the expected proportion $\gamma n$, where $n$ is the total number of tokens in the sequence. A significantly high $z$-score yields a small p-value, indicating the presence of the watermark.

**Gumbel Watermark.** The watermarking techniques proposed by Aaronson (2023) and Kuditipudi et al. (2023) can be described using a sampling algorithm based on the Gumbel trick (Zhao et al., 2024). This algorithm hashes the preceding $h$ tokens using the key k to obtain a score $r_i$ for each token $i$ in the vocabulary $\mathcal{V}$, where each $r_i$ is uniformly distributed in $[0, 1]$. The next token is chosen deterministically as follows: $\arg\max_{y_i \in \mathcal{V}} [\log P(y_i | x_{<t}) - \log(-\log(r_{y_i}))]$. Thus, given a random vector $r \sim (\text{Uniform}([0, 1]))^{|\mathcal{V}|}$, $-\log(-\log(r_{y_i}))$ follows a Gumbel(0,1) distribution. This results in a distortion-free deterministic sampling algorithm (for large $h$) for generating text. During detection, if the observed score

$$\text{Score}(y) = \sum_{t=1}^{n} \log\left(1/(1 - r_{y_t})\right) \tag{2}$$

is high, the p-value is low, indicating the presence of the watermark.

## 3 METHOD

**Problem Statement** Identifying watermark segments within a long text sequence $y$ presents two key challenges. First, we need to design a classification rule $\mathcal{M}(x) \rightarrow \{0, 1\}$ that determines whether $y$ contains a watermark segment. To address this, we propose the *Geometric Cover Detector* (GCD), which enables multi-scale watermark detection. Second, accurately locating the watermark segments $y_{s_i:e_i}$ within the full sequence $y$ requires finding the start and end token indices, $s_i$ and $e_i$, for each watermark segment. We introduce the *Adaptive Online Locator* (AOL) with the Aligator algorithm to precisely identify the position of the watermarked text span within the longer sequence.

---

**Algorithm 1** Geometry Cover Watermark Detection

---

**Input:** Mixed-source text $y$ of length $n$, target false positive rate (FPR) $\tau$, watermark detector score function Score, FPR calibration function $F$

1: Divide $y$ into Geometry Cover set $\mathcal{I}$ as defined in Equation 3
2: **for** each interval $\mathcal{I}_t : (i_t, j_t)$ in Geometry Cover set **do**
3:     Compute FPR $\alpha \leftarrow F(y_{i_t:j_t}, \text{Score}(y_{i_t:j_t}))$
4:     **if** $\alpha < \tau$ **then**
5:         **return** 1, i.e., "The sequence contains a watermark"
6: **return** 0, i.e., "No watermark found"

---

### 3.1 WATERMARK SEGMENT CLASSIFICATION

A straightforward approach to detect whether an article contains watermarked text is to pass it through the original watermark detector (as we discussed in Section 2). If the detection score from the original detector is larger than a threshold, the text contains a watermark; otherwise, no watermark is found. However, this approach is ineffective for long, mixed-source texts where only a small portion originates from the watermarked LLM. Since a large portion of the text lacks the watermark signal, the overall score for the entire document will be dominated by the unwatermarked portion, rendering the detection unreliable.

To overcome this limitation, we need a method that analyzes the text at different scales or chunks. If a chunk is flagged as watermarked, we can then classify the entire sequence as containing watermarked text. The question then becomes: how do we design these intervals or chunks effectively? We leverage the Geometric Cover (GC) technique introduced by Daniely et al. (2015) to construct an efficient collection of intervals for analysis.

Geometric Cover (GC) is a collection of intervals belonging to the set $\mathbb{N}$, defined as follows:

$$\mathcal{I} = \bigcup_{k \in \mathbb{N} \cup 0} \mathcal{I}^{(k)}, \text{ where } \forall k \in \mathbb{N} \cup 0, \text{ and } \mathcal{I}^{(k)} = [i \cdot 2^k, (i+1) \cdot 2^k - 1] : i \in \mathbb{N}. \quad (3)$$

Essentially, each $\mathcal{I}^{(k)}$ represents a partition of $\mathbb{N}$ into consecutive intervals of length $2^k$. For example, $\mathcal{I}^{(4)}$ contains all consecutive 16-token intervals. Due to this structure, each token belongs to $\lfloor \log n \rfloor + 1$ different intervals (as illustrated in Figure 1), and there are a total of $n + n/2 + n/4 + n/8 + \cdots = \mathcal{O}(n)$ intervals in the GC set. This allows us to establish a multi-scale watermark detection framework. Moreover, Lemma 5 from Daniely et al. (2015) ensures that for any unknown watermarked interval, there is a corresponding interval in the geometric cover that is fully contained within it and is at least *one-fourth* its length. This ensures the effectiveness of watermark detection using the geometric cover framework.

Leveraging the GC construction, our multi-scale watermark detection framework divides the input text into segments based on the GC intervals. In real-world applications, we need to balance the granularity of the intervals. For instance, classifying a 4-token chunk as watermarked might not be convincing. Therefore, we start from higher-order intervals, such as $\mathcal{I}^{(5)}$, which comprises all geometric cover intervals longer than 32 tokens.

Algorithm 1 outlines our approach. For each segment $\mathcal{I}_t : y_{i_t:j_t}$ in the GC, we first compute a detection score using the appropriate watermark detector for the scheme employed (e.g., Equation 1 for Red-Green Watermark or Equation 2 for Gumbel Watermark). This score, along with the segment itself, is then passed to an FPR calibration function $F$. This function estimates the FPR associated with the segment. Further details on FPR calibration can be found in the Appendix A.2.

If the estimated FPR, denoted as $\alpha$, falls below a predefined target FPR ($\tau$), we classify the entire sequence as containing a watermark. It is important to note that $\tau$ is set at the segment level. Using the union bound, consider a mixed-source text composed of $n$ tokens. The geometric cover of the text is constructed from $\mathcal{O}(n)$ intervals. Let $\tau$ represent the false positive rate for each interval test (Type I error rate). In this case, the Family-Wise Error Rate (FWER), which is the probability of incorrectly classifying the entire document as watermarked, is bounded by $n\tau$.

### 3.2 PRECISE WATERMARK POSITION LOCALIZATION

---

**Algorithm 2** Watermark Position Localization

---

**Input:** Mixed-source text $y$, threshold $\zeta$, iterations $m$

1: Get watermark detection scores of each token for $y$ from watermark detector $\{s_t\}_{t \in [n]}$
2: Initialize Aligator algorithm $\mathcal{A}$ with circular starting strategy
3: **for** $i = 1$ to $m$ **do**
4:     Random starting position $k \leftarrow$ random index in $\{1, \ldots, n\}$
5:     Predict the pointwise estimate of the expected detection score for each token in the $i$-th round:

$$\boldsymbol{\theta}^{(i)} := \{\theta_t\}_{t \in [n]}^{(i)} \leftarrow \mathcal{A}(s_k, s_{k+1}, \ldots, s_n, s_1, \ldots, s_{k-1})$$

6: **end for**
7: Average predicted scores across all rounds $\overline{\boldsymbol{\theta}} \leftarrow \frac{1}{m} \sum_{i=1}^{m} \boldsymbol{\theta}^{(i)}$
8: Identify watermarked positions $\mathcal{W} \leftarrow \{t \mid \overline{\boldsymbol{\theta}}_t > \zeta\}$
9: **return** $\mathcal{W}$

---

While the previous section focused on detecting the presence of watermarks, simply knowing a watermark exists doesn't reveal which specific paragraphs warrant scrutiny. Here, we aim to localize the exact location of watermarked text. A naive approach would involve iterating through all possible interval combinations within the sequence, applying the watermark detection rule to each segment $y_{i:j}$ for all $i \in \{1, \ldots, n\}$ and $j \in \{i, \ldots, n\}$. While this brute-force method can identify watermark segments, its $\mathcal{O}(n^2)$ time complexity makes it computationally expensive for long sequences.

Furthermore, relying solely on individual token scores for localization is unreliable due to the inherent noise in the watermarking process. To address this issue, we propose to formulate it as a **sequence denoising problem** (a.k.a., smoothing or nonparametric regression) so we can provide a pointwise estimate of the *expected* detection score *for each token*. Specifically, the denoising algorithm tasks a sequence of noisy observations $s_1, \ldots, s_n$ and output $\{\theta_t\}_{t \in [n]}$ as an estimate to $\{\mathbb{E}[s_t]\}_{t \in [n]}$.

As an example, for the Green-Red Watermark, the sequence of noisy observations $\{s_t = \mathbf{1}(y_t \in \text{Green Tokens})\}_{t \in [n]}$ consists of Bernoulli random variables. The expectation $\mathbb{E}[s_t] = \gamma$ if $y_t$ is not watermarked and $\mathbb{E}[s_t] > \gamma$ otherwise. For the Gumbel Watermark, the noisy observations $\{s_t = \log(1/(1 - r_{y_t}))\}_{t \in [n]}$ consists of exponential random variables satisfying $\mathbb{E}[s_t] = 1$ if $y_t$ is unwatermarked and larger otherwise. The intuition is that, while individually they are too noisy, if we average them appropriately within a local neighborhood, we can substantially reduce the noise. If we can accurately estimate the sequence $\mathbb{E}[s_i]$, we can localize watermarked segments by simply thresholding the estimated score pointwise.

The challenge, again, is that we do not know the appropriate window size to use. In fact, the appropriate size of the window should be larger if $s_i$ is in the interior of a long segment of either watermarked or unwatermarked text. The sharp toggles among text from different sources add additional challenges to most smoothing algorithms.

For these reasons, we employ the Aligator (**A**ggregation of on**LI**ne avera**G**es using **A** geome**T**ric c**O**ve**R**) algorithm (Baby et al., 2021). In short, Aligator is an online smoothing algorithm that optimally competes with an oracle that knows the segments of watermarked sequences ahead of time. The algorithm employs a Geometric Cover approach internally, where words positioned mid-paragraph are typically included in multiple intervals of varying lengths for updates. Notably, Aligator provides the following estimation guarantee:

$$\frac{1}{n} \sum_t (\theta_t - \mathbb{E}[s_t])^2 = \tilde{O}\left(\min\left\{n^{-1}(1 + \sum_{t=2}^{n} \mathbf{1}_{\mathbb{E}[s_t] \neq \mathbb{E}[s_{t-1}]}), n^{-1} \vee n^{-2/3}(\sum_{t=2}^{n} |\mathbb{E}[s_t] - \mathbb{E}[s_{t-1}]|)\right\}\right).$$

Moreover, for all segments with start/end indices $(i, j) \in [n]^2$, i.e.

$$\frac{1}{j-1} \sum_{t=i}^{j} (\theta_t - \frac{1}{j-i} \sum_{t'=i}^{j} \mathbb{E}[s_{t'}])^2 \leq \tilde{O}(1/\sqrt{j-i}).$$

This ensures that for every segment, the estimated value is as accurate as statistically permitted. The time complexity for Aligator is $\mathcal{O}(n \log n)$. For a detailed implementation of Aligator, please refer to the original paper (Baby et al., 2021). For the theoretical results, see (Baby & Wang, 2021).

**Circular Aligator.** To mitigate the boundary effects common in online learning, where prediction accuracy suffers at the beginning and end of sequences, we introduce a circular starting strategy. Instead of processing the text linearly, we treat it as a circular buffer. For each iteration, we randomly choose a starting point and traverse the entire sequence, effectively mitigating edge effects. The final prediction for each token is then obtained by averaging the predictions across all iterations.

Finally, we apply a threshold to this denoised average score function to delineate the boundaries of watermark segments within the text (as illustrated in Figure 1). The high-level implementation of this method is detailed in Algorithm 2. This approach enables us to precisely identify the location of suspected plagiarism within large documents with high confidence, facilitating further investigation and verification.

## 4  EXPERIMENT

**Datasets and Mixed-source Texts**   We utilize two text datasets: C4 (Raffel et al., 2020) and Arxiv (Cohan et al., 2018). The "Colossal Clean Crawled Corpus" (C4) dataset is a collection of English-language text sourced from the public Common Crawl web scrape, a rich source for unwatermarked human-written text. We use random samples from the news-like subset of the C4 dataset in our experiments. The Arxiv dataset is part of the Scientific-Papers dataset collected from scientific repositories, arXiv.org and PubMed.com. We use the Arxiv split in our experiments, which contains abstracts and articles of scientific papers. Both datasets are used to construct watermarked positive samples and human-written negative samples. To transform unwatermarked samples into partially watermarked samples, we randomly select 3-5 sentences in a long text and set them as prompts. Then, we generate 300 tokens of watermarked text conditioned on the prompts using large language models. The generated responses replace the original suffix sentences after the prompt. In this way, we embed 300-token watermarks into 3000-token contexts from the datasets, making the watermark 10% of the mix-sourced text. We randomly choose the position of the watermark in this longer context and record the locations for later testing. Our goal is to determine if a document contains watermark text and locate its position. For each dataset, we use 500 samples as the test set to show the results.

**Language Models and Watermarking Methods.**   We use the publicly available LLaMA-7B (Touvron et al., 2023) and Mistral (Jiang et al., 2023) models. To verify the general applicability of the watermark detection methods, we select three watermarking techniques: Gumbel-Watermark (Aaronson, 2023), KGW-Watermark (Kirchenbauer et al., 2023), and Unigram-Watermark (Zhao et al., 2023). These methods represent the state-of-the-art watermarking approaches for large language models, offering high quality, detectability, and robustness against adversarial attacks. For all watermarking generations, we configure the temperature to 1.0 for multinomial sampling. Additionally, for KGW-Watermark and Unigram-Watermark, we set the green token ratio $\gamma$ to 0.5 and the perturbation $\delta$ to 2.0.

**Baselines**   In watermark segment detection, we use the original watermark detector in each watermarking method as the VANILLA baseline to compare with our approach GCD. In watermark segment localization, we use RoBERTa (Liu et al., 2019) models for comparing with our method AOL. We train each RoBERTa (designed for different watermarking methods) to predict whether a sequence is a watermarked sequence or not, given the watermark detection scores $r$ for each token. We add an extra fully connected layer after getting the representation of the [CLS] token. We construct 1000 training samples with 60 token scores as input and the binary label of this segment as the label. We train the RoBERTa model for 20 epochs and enable early stopping if the loss converges. It can reach over 90% accuracy in the training set. During testing on mixed-source text, we employ the sliding window idea to test each chunk for watermarks and then calculate the IoU score.

**Evaluation**   For the watermarked text classification task, we report the true positive rates (TPR) based on different specified false positive rates (FPR). Maintaining a low FPR is critical to ensure that human-written text is rarely misclassified as LLM-generated text.

Since the FPR at the per-instance level differs from the document-level FPR, we calibrate FPR to three distinct levels in each scenario to enable fair comparisons. Specifically, we manipulate the pre-segment FPR (SEG-FPR) by adjusting the threshold parameter $\tau$ as outlined in Algorithm 1. Then, we can get the empirical document FPR (DOC-FPR) by evaluating our method GCD based on

Table 1: True Positive Rate (TPR) at various False Positive Rate (FPR) levels for baseline VANILLA and our method GCD. For each setting, we select three distinct segment-level FPRs (SEG-FPR) and compare the performance of VANILLA and GCD at equivalent document-level FPRs (DOC-FPR). GCD consistently outperforms VANILLA across different models and datasets.

| Method | KGW-Watermark TPR | | | Unigram-Watermark TPR | | | Gumbel-Watermark TPR | | |
|---|---|---|---|---|---|---|---|---|---|
| *C4 Dataset, Llama-7B* | | | | | | | | | |
| SEG-FPR | 1e-5 | 5e-5 | 1e-4 | 1e-4 | 2e-4 | 0.001 | 1e-4 | 0.001 | 0.010 |
| DOC-FPR | 0.034 | 0.076 | 0.082 | 0.002 | 0.004 | 0.030 | 0.026 | 0.080 | 0.358 |
| VANILLA | 0.602 | 0.676 | 0.692 | 0.006 | 0.006 | 0.058 | 0.650 | 0.762 | 0.918 |
| GCD | **0.912** | **0.934** | **0.934** | **0.874** | **0.906** | **0.958** | **1.000** | **1.000** | **1.000** |
| *C4 Dataset, Mistral-7B* | | | | | | | | | |
| SEG-FPR | 1e-5 | 1e-4 | 2e-4 | 0.001 | 0.010 | 0.020 | 1e-4 | 5e-4 | 0.001 |
| DOC-FPR | 0.037 | 0.087 | 0.153 | 0.001 | 0.012 | 0.040 | 0.024 | 0.046 | 0.054 |
| VANILLA | 0.697 | 0.830 | 0.877 | 0.000 | 0.012 | 0.030 | 0.690 | 0.760 | 0.780 |
| GCD | **0.960** | **0.983** | **0.990** | **0.722** | **0.974** | **1.000** | **0.970** | **0.980** | **0.990** |
| *Arxiv Dataset, Llama-7B* | | | | | | | | | |
| SEG-FPR | 1e-5 | 5e-5 | 2e-4 | 1e-4 | 2e-4 | 0.001 | 1e-4 | 0.001 | 0.010 |
| DOC-FPR | 0.068 | 0.116 | 0.186 | 1e-4 | 2e-4 | 0.014 | 0.024 | 0.066 | 0.280 |
| VANILLA | 0.844 | 0.896 | 0.908 | 0.000 | 0.000 | 0.026 | 0.593 | 0.655 | 0.823 |
| GCD | **0.990** | **0.994** | **0.996** | **0.892** | **0.922** | **0.974** | **0.958** | **0.978** | **1.000** |
| *Arxiv Dataset, Mistral-7B* | | | | | | | | | |
| SEG-FPR | 1e-5 | 1e-4 | 2e-4 | 0.001 | 0.020 | 0.020 | 1e-5 | 1e-4 | 2e-4 |
| DOC-FPR | 0.033 | 0.197 | 0.253 | 0.001 | 0.028 | 0.036 | 0.082 | 0.192 | 0.230 |
| VANILLA | 0.757 | 0.883 | 0.907 | 0.002 | 0.032 | 0.088 | 0.860 | 0.930 | 0.930 |
| GCD | **0.967** | **0.990** | **1.000** | **0.566** | **0.920** | **0.964** | **0.950** | **0.960** | **0.970** |

pure natural text. For VANILLA, we set the FPR according to GCD's empirical FPR and subsequently test for its empirical TPR.

For locating specific watermark segments, we calculate the Intersection over Union (IoU) score to measure the accuracy of watermark segment localization. The IoU score computes the ratio of the intersection and union between the ground truth and inference, serving as one of the main metrics for evaluating the accuracy of object detection algorithms:

$$IoU = \frac{\text{Area of Intersection}}{\text{Area of Union}} = \frac{|\text{Detected Tokens} \cap \text{Watermarked Tokens}|}{|\text{Detected Tokens} \cup \text{Watermarked Tokens}|}$$

## 4.1 DETECTION RESULTS

**Watermark Segment Classification Results**  As shown in Table 1, our proposed *Geometric Cover Detector* (GCD) consistently outperforms the baseline VANILLA method across all watermarking techniques and large language models on both the C4 and Arxiv datasets. The robustness of GCD across diverse conditions underscores its effectiveness in watermark segment classification, demonstrating clear superiority over VANILLA. Additionally, we observe that VANILLA exhibits near-zero detection rates when the target false positive rate is low. This suggests that VANILLA struggles to detect watermarked segments in longer contexts, as the watermark signal weakens, rendering the simpler detector ineffective.

**Precise Watermark Position Localization Results**  For the watermark position localization task, we evaluate our proposed method AOL against the baseline method ROBERTA (Table 2). We calculate the average IoU score to quantify the precision of the watermark localization. Our method consistently outperforms the baseline across all test settings. For example, on the C4 dataset using the mistral-7B model, AOL achieves a substantially higher IoU score of 0.809 compared to 0.301 for ROBERTA. We also test AOL's ability to detect multiple watermarks by inserting 3x300-token Gumbel watermarks (generated by Mistral-7B) into 6000-token texts. Across 200 samples, the average IoU for detecting the watermarks is 0.802, demonstrating AOL's effectiveness for multiple watermark detection. Figure 2 provides a case example illustrating the improved localization performance of AOL on the Gumbel watermark with the Mistral-7B model. The upper image shows the boundary effects of using online learning. The lower image demonstrates more precise localization resulting from the circular starting strategy with 10 random starting points.

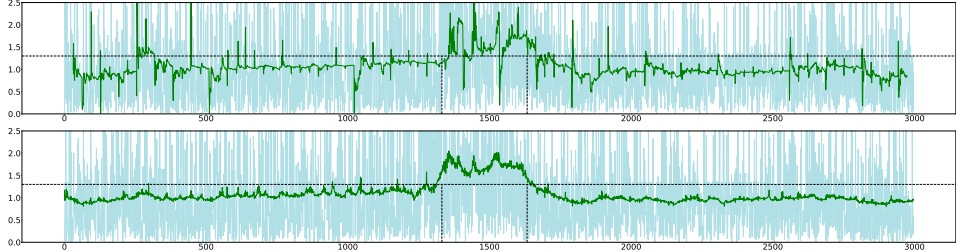

Figure 2: Example of precise watermark localization using AOL with Gumbel Watermark. Light green lines show token scores, and dark green lines show predicted mean scores. The horizontal dashed line shows the score threshold $\zeta = 1.3$. The vertical dashed line marks the original watermark position. The top image demonstrates inaccurate localization from a single pass of the Aligator algorithm, highlighting boundary artifacts. In contrast, the bottom image shows precise localization achieved by AOL's circular initialization strategy with $m = 10$ random starts.

Table 3: VANILLA and GCD watermark segment classification result with Unigram Watermark on Mistral-7B with different target false positive rates.

| Length | Method | TPR | | |
|---|---|---|---|---|
| | | FPR-1 | FPR-2 | FPR-3 |
| 3000 | VANILLA | 0.000 | 0.012 | 0.038 |
| | GCD | **0.722** | **0.974** | **1.000** |
| 6000 | VANILLA | 0.000 | 0.000 | 0.005 |
| | GCD | **0.730** | **0.980** | **1.000** |
| 9000 | VANILLA | 0.000 | 0.000 | 0.000 |
| | GCD | **0.730** | **0.980** | **1.000** |
| 18000 | VANILLA | 0.000 | 0.000 | 0.000 |
| | GCD | **0.730** | **0.980** | **1.000** |

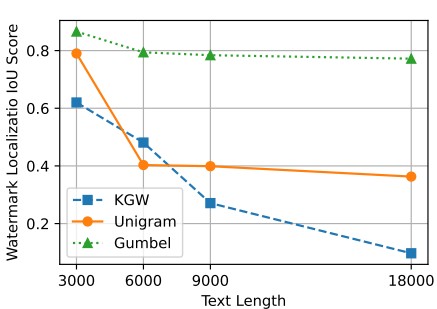

Figure 3: Watermark localization results using different watermarking methods and varying text lengths.

## 4.2 DETECTION RESULTS WITH DIFFERENT LENGTHS

As mentioned previously, watermark detection can easily be disturbed by long natural paragraphs, and our approach aims to minimize the effect of length scale. We test our method on texts of varying total lengths, ranging from 3000 to 18000 tokens, while keeping the watermark segment length constant at 300 tokens. The same detection threshold and parameters used for 3000 total tokens are applied across all lengths. We find that the Gumbel watermark segment classification performs well even as total length increases, as shown in Table 3. For repetitive watermarks like KGW and Unigram, longer texts in the Geometry Cover also cause a decrease in segment detection, as shown in Figure 3. However, compared

Table 2: Precise Watermark Position Localization Performance: Intersection over Union (IoU) score for baseline ROBERTA and our method AOL. AOL consistently outperforms ROBERTA.

| Method | KGW-WM IoU | Unigram-WM IoU | Gumbel-WM IoU |
|---|---|---|---|
| *C4 Dataset, Llama-7B* | | | |
| ROBERTA | 0.563 | 0.444 | 0.535 |
| AOL | **0.657** | **0.818** | **0.758** |
| *C4 Dataset, Mistral-7B* | | | |
| ROBERTA | 0.238 | 0.019 | 0.301 |
| AOL | **0.620** | **0.790** | **0.809** |
| *Arxiv Dataset, Llama-7B* | | | |
| ROBERTA | 0.321 | 0.519 | 0.579 |
| AOL | **0.718** | **0.862** | **0.635** |
| *Arxiv Dataset, Mistral-7B* | | | |
| ROBERTA | 0.372 | 0.249 | 0.421 |
| AOL | **0.571** | **0.682** | **0.802** |

to directly detecting on the whole paragraph, this decrease is more acceptable. Importantly, the parameters used in these tests are identical to those for 3000 tokens. In practice though, for texts of different lengths, the number of starting points in the circular buffer should be adjusted accordingly. This way, similarly strong results can be achieved as with 3000 tokens.

## 5 CONCLUSION

This paper introduces novel methods for partial watermark detection in LLM-generated text, addressing the critical need for identifying watermark segments within longer, mixed-source documents. By leveraging the geometric cover trick and the Alligator algorithm, our approach achieves high accuracy in both classifying and localizing watermarks, significantly outperforming baseline methods. These advancements pave the way for more robust and reliable detection of synthetic text, promoting responsible use and mitigating potential misuse of LLMs in various domains.

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

# A  APPENDIX

## A.1  MORE ON RELATED WORK

**Text Attribution and Plagiarism Detection**  Watermark text localization shares similarities with text attribution and plagiarism detection, particularly in the aspect of pinpointing specific text segments. Commercial plagiarism detection systems like Turnitin, Chegg, and Grammarly rely on vast databases to identify copied content, highlighting similar segments. Research in plagiarism localization, such as the work by Grozea et al. (2009), focuses on precisely identifying copied passages within documents. Their approach utilizes a similarity matrix and sequence-matching techniques for accurate localization. Similarly, the "Greedy String Tiling" algorithm (Wise, 1996) has been successfully employed by Mozgovoy et al. (2010) for identifying overlapping text. However, these methods require reference files in a database, whereas watermark text localization aims to localize the watermark text using a watermark key, eliminating the need for reference documents. Detecting partially watermarked text presents a unique challenge, akin to an online learning problem, where tokens in watermark segments exhibit special signals that can be captured by a strongly adaptive online learning algorithm like Aligator (Baby et al., 2021).

**Identifying Watermarked Portions in Long Text**  To detect watermarked portions in long texts, Aaronson (2023) designs a "watermark plausibility score" for each interval. Given $\{s_t = \log(1/(1 - r_{y_t}))\}_{t \in [n]}$, the watermark plausibility score is $\frac{(\sum_{t=i}^{j} s_t)^2}{j-i} - L$, where $L$ is a constant. This method draws connections to change point detection algorithms, aiming to maximize the sum of plausibility scores to detect watermarked portions. Aaronson (2023) manages to reduce the time complexity from $\mathcal{O}(n^2)$ to $\mathcal{O}(n^{3/2})$. Additionally, Christ et al. (2023) demonstrate how to detect a watermarked contiguous substring of the output with sufficiently high entropy, calling the algorithm *Substring Completeness*. This algorithm has a time complexity of $\mathcal{O}(n^2)$. A recent, independent work of Kirchenbauer et al. (2024) introduces the *WinMax* algorithm to detect watermarked sub-regions in long texts. This algorithm searches for the continuous span of tokens that produces the highest $z$-score by iterating over all possible window sizes and traversing the entire text for each size, with a time complexity of $\tilde{\mathcal{O}}(n^2)$. Our *Adaptive Online Locator* (AOL) improves the efficiency of detecting watermarked portions, reducing the time complexity to $\mathcal{O}(n \log n)$.

## A.2  FPR CALIBRATION FUNCTION $F$

As discussed in Section 3.1, the FPR Calibration Function calculates the p-value / FPR for per-instance watermark detection, given the detection scores and the original text. We follow the methodologies outlined in Zhao et al. (2023) and Fernandez et al. (2023) for FPR calibration. This section presents three methods for detecting KGW-Watermark, Unigram-Watermark, and Gumbel-Watermark, each employing a unique scoring mechanism and statistical test to assess the FPR.

### A.2.1  KGW-WATERMARK

For the KGW-Watermark scheme described in Kirchenbauer et al. (2023), we follow the approach in Fernandez et al. (2023). When detecting the watermark for a text segment, under the null hypothesis $\mathcal{H}_0$ (i.e., the text is not watermarked), the score $\text{Score}(y) = \sum_{t=1}^{n} \mathbf{1}(y_t \in \text{Green Tokens})$ follows a binomial distribution $\mathcal{B}(n, \gamma)$, where $n$ is the total number of tokens and $\gamma$ is the probability of a token being part of the green list. The p-value for an observed score $s$ is calculated as:

$$\text{p-value}(s) = \mathbb{P}(\text{Score}(y) > s \mid \mathcal{H}_0) = I_\gamma(s, n - s + 1),$$

where $I_x(a, b)$ is the regularized incomplete Beta function.

### A.2.2  UNIGRAM-WATERMARK

For the Unigram-Watermark scheme, we adopt the methodologies from Zhao et al. (2023). To achieve a better FPR rate, the detection score differs from the KGW-Watermark approach. The score is defined as $\text{Score}(y) = \sum_{t=1}^{m} \mathbf{1}(\tilde{y}_t \in \text{Green Tokens})$, where $\tilde{y} = \text{Unique}(y)$ represents the sequence of unique tokens in text $y$, and $m$ is the number of unique tokens.

Under the null hypothesis $\mathcal{H}_0$ (i.e., the text is not watermarked), each token has a probability $\gamma$ of being included. Using the variance formula for sampling without replacement ($N$ choose $\gamma N$), the variance of this distribution is:

$$\text{Var}\left[\sum_{t=1}^{m} \mathbf{1}(\tilde{y}_t \in \text{Green Tokens}) \mid y\right] = m\gamma(1-\gamma)(1 - \frac{m-1}{n-1}),$$

where $n$ is the total number of tokens, and $\gamma$ is the probability of a token being in the green list. The conditional variance of $z_{\text{Unique}(y)}$ is thus $(1 - \frac{m-1}{n-1})$. The false positive rate (FPR) is then given by:

$$\text{FPR} = 1 - \Phi\left(\frac{z_{\text{Unique}(y)}}{\sqrt{1 - \frac{m-1}{n-1}}}\right),$$

where $\Phi$ is the standard normal cumulative distribution function.

### A.2.3 GUMBEL WATERMARK

For the Gumbel Watermark (Aaronson, 2023), we adopt the approach presented in Fernandez et al. (2023), which utilizes a gamma test for watermark detection. Under the null hypothesis $\mathcal{H}_0$, $\text{Score}(y) = \sum_{t=1}^{n} \log\left(1/(1 - r_{y_t})\right)$ follows a gamma distribution $\Gamma(n, 1)$. The p-value for an observed score $s$ is calculated as:

$$\text{p-value}(s) = \mathbb{P}(\text{Score}(y) > s \mid \mathcal{H}_0) = \frac{\Gamma(n, s)}{\Gamma(n)}$$

where $\Gamma(n, s)$ is the upper incomplete gamma function and $n$ is the total number of tokens.

For all three methods, a lower p-value indicates stronger evidence against the null hypothesis, suggesting a higher likelihood that the text is watermarked. These methods provide a comprehensive framework for watermark detection, each offering unique advantages depending on the specific characteristics of the text and the desired sensitivity of the detection process.

### A.3 DETECTION ROBUSTNESS AGAINST ATTACKS

Table 4: Watermark segment classification and localization performance with different attacks.

| Method | KGW-Watermark TPR and IoU | | | | Unigram-Watermark TPR and IoU | | | | Gumbel-Watermark TPR and IoU | | | |
|---|---|---|---|---|---|---|---|---|---|---|---|---|
| | FPR-1 | FPR-2 | FPR-3 | AOL IoU | FPR-1 | FPR-2 | FPR-3 | AOL IoU | FPR-1 | FPR-2 | FPR-3 | AOL IoU |
| *Random Swap* | | | | | | | | | | | | |
| Baseline | 0.190 | 0.340 | 0.460 | – | 0.000 | 0.005 | 0.025 | – | 0.110 | 0.150 | 0.160 | – |
| Ours | 0.175 | 0.325 | 0.380 | 0.095 | 0.740 | 0.990 | 1.000 | 0.472 | 0.390 | 0.550 | 0.560 | 0.325 |
| *Random Delete* | | | | | | | | | | | | |
| Baseline | 0.310 | 0.440 | 0.545 | – | 0.000 | 0.000 | 0.015 | – | 0.255 | 0.300 | 0.325 | – |
| Ours | 0.645 | 0.750 | 0.820 | 0.269 | 0.630 | 0.905 | 0.960 | 0.475 | 0.750 | 0.830 | 0.850 | 0.613 |
| *ChatGPT Paraphrase* | | | | | | | | | | | | |
| Baseline | 0.050 | 0.195 | 0.335 | – | 0.000 | 0.000 | 0.005 | – | 0.020 | 0.065 | 0.065 | – |
| Ours | 0.050 | 0.100 | 0.165 | 0.032 | 0.040 | 0.145 | 0.510 | 0.218 | 0.075 | 0.110 | 0.130 | 0.090 |

We evaluate the robustness of our watermark detection method against three types of attacks (Table 4). First, we use GPT-3.5-turbo to rewrite the text segments containing the watermark as the paraphrasing attack. The other two attacks randomly swap or delete words at a ratio of 0.2. As expected, rewriting by ChatGPT is the most damaging attack, leading to a decline in detection performance. However, our detection method still significantly outperforms the baseline direct detection across most attack types in terms of TPR. For watermark localization, measured by intersection over union (IoU), our method still generates satisfactory results under these attacks. Overall, the results demonstrate the robustness of our watermark detection approach against various perturbations to the watermarked text.

### A.4 DISCUSSION AND LIMITATION

Our methods, GCD and AOL, can be applied to other watermarking schemes as long as they have token-wise detection scores for the sequence, such as Hu et al. (2024) and Zhao et al. (2024). The

detection results are constrained by the strength of the original watermark generation and the quality of the prompt text. In some cases, low-quality text produced by the watermark generation method cannot be directly detected using the original detection method. Additionally, positive samples created by inserting the generated watermark paragraph into natural text may not be detectable with our approach. However, these limitations arise from the current limitations of watermark generation and detection methods themselves, which is outside the scope of detecting small watermarked segments within long text, the focus of this work. Therefore, we assume that our method needs only to detect reasonably high quality watermarked text segments embedded in long text.

## A.5 DATA FILTERING AND HYPERPARAMETERS

We first extract random consecutive sentences from the text as prompts (A), generate watermarked continuation sentences (B) using the language model, and insert B after A to create a partially watermarked text. However, due to limitations of the watermarking method, the quality of some generated segments (B) is poor, and even direct detection cannot accurately predict the watermark. In such cases, we check the watermark quality during generation and remove segments B with poor-quality watermarks based on the p-value for the KGW and Gumbel methods, and the z-score for the Unigram method. Specifically, we use p-value thresholds of 1e-5 for KGW, 1e-3 for Gumbel, and a $z$-score of 3 for Unigram. For precise watermark localization, we use 30 starting points in the circular buffer, primarily considering the total text length. This hyperparameter can be dynamically adjusted based on the text length to balance the smoothness of boundary detection versus interior detection and computation cost. More starting points bring edge detection closer to interior detection but increase computation cost. Thus, the number of starting points involves a trade-off between these factors.

