# OpenReview forum: "Efficiently Identifying Watermarked Segments in Mixed-Source Texts"
_NeurIPS.cc/2024/Workshop/SafeGenAi — SafeGenAi Poster_

### Official Review · Reviewer_7nRo · 2024-10-09
**The authors present a novel watermarking detection technique that can detect watermarked segments of a large body of text**

**Rating:** 7
**Confidence:** 4

**Review:**

Pros:
1) The online aggregation algorithm for consolidating the results of a large number of p-values is compelling, especially for this use case.
2) The use cases for this technique are practical and discussed at length in the paper.

Cons:
1) The paper can be decently difficult to read, with a large amount of implementation details/theoretical complexity that could be left to the appendix to significantly improve readability.

---

### Official Review · Reviewer_wVHr · 2024-10-12
**Mixed-Source Watermarking**

**Rating:** 9
**Confidence:** 4

**Review:**

This is an excellent write-up in which the authors present their novel approaches to watermark detection, advancing beyond simple text-level classification to identifying watermark segments within long, mixed-source texts. They effectively utilize the geometric cover trick and the Alligator algorithm from online learning to detect and localize watermark segments with both efficiency and accuracy. While the results are impressive, I would like to clarify one point. Existing watermarks, particularly the state-of-the-art methods like KGW and Unigram, are not limited to text-level classification. Their detection algorithms can also identify watermarks in mixed-source texts; however, they may not perform as well as this novel method, which specifically targets that challenge.